# Cross-GWAS coherence test at the gene and pathway level

**Daniel Krefl** [1,2] *, **Sven Bergmann** [1,2,3] *

**1** Department of Computational Biology, University of Lausanne, Lausanne, Switzerland, **2** Swiss Institute of Bioinformatics, Lausanne, Switzerland, **3** Dept. of Integrative Biomedical Sciences, University of Cape Town, Cape Town, South Africa

* daniel.krefl@unil.ch (DK); sven.bergmann@unil.ch (SB)

**Data Availability Statement:** Gene annotation can be downloaded from Ensemble BioMart https://www.ensembl.org/ (we used release 104). 1K Genome project reference panel can be obtained from https://www.internationalgenome.org/ (we used the 30x high coverage GRCh38 release). The

## Abstract

Proximal genetic variants are frequently correlated, implying that the corresponding effect sizes detected by genome-wide association studies (GWAS) are also not independent. Methods already exist to account for this when aggregating effects from a single GWAS across genes or pathways. Here we present a rigorous yet fast method for detecting genes with coherent association signals for two traits, facilitating cross-GWAS analyses. To this end, we devised a new significance test for the covariance of datapoints not drawn independently but with a known inter-sample covariance structure. We show that the distribution of its test statistic is a linear combination of $\chi^2$ distributions with positive and negative coefficients. The corresponding cumulative distribution function can be efficiently calculated with Davies' algorithm at high precision. We apply this general framework to test for dependence between SNP-wise effect sizes of two GWAS at the gene level. We extend this test to detect also gene-wise causal links. We demonstrate the utility of our method by uncovering potential shared genetic links between the severity of COVID-19 and (1) being prescribed class M05B medication (drugs affecting bone structure and mineralization), (2) rheumatoid arthritis, (3) vitamin D (25OHD), and (4) serum calcium concentrations. Our method detects a potential role played by chemokine receptor genes linked to $T_H1$ versus $T_H2$ immune response, a gene related to integrin beta-1 cell surface expression, and other genes potentially impacting the severity of COVID-19. Our approach will be useful for similar analyses involving datapoints with known auto-correlation structures.

## Author summary

Genome-wide association studies (GWAS) deliver effect size estimates of a given trait for millions of Single Nucleotide Polymorphisms (SNPs). Powerful tools already exist using these summary statistics to elucidate the global joint genetic contribution to a pair of traits, such as *cross-trait LD-score regression*, but these methods cannot reveal the joint contributions at the level of genes and pathways. Here we present a novel methodology to co-analyze the association data from a pair of GWAS to identify genes and pathways that may be relevant to both individual traits. Our test novelty is that a gene is considered co-

MSigDB v7.4 database used for pathway enrichment tests can be downloaded at https://www.gsea-msigdb.org/gsea/msigdb/. The used GWAS summary statistics are published by the authors of the cited original studies on the following websites. BMDs: http://www.gefos.org/?q=content/data-release-2018 Calcium: https://gwas.mrcieu.ac.uk/datasets/ukb-d-30680_irnt/ COVID-19: https://www.covid19hg.org/results/r5/ (A2 ALL eur leave ukbb 23andme, release 7. Jan. 2021) Drug classes: https://cnsgenomics.com/content/data Estradiol and Estrone: http://www.gefos.org/?q=content/estrogen-gwas-2018 RA: https://plaza.umin.ac.jp/~yokada/datasource/software.htm Vitamin D: https://www.ebi.ac.uk/gwas/efotraits/EFO_0004631 Code availability: The used methods have been implemented in the python package PascalX (version 0.0.2), available on GitHub (https://github.com/BergmannLab/PascalX) and Zenodo https://doi.org/10.5281/zenodo.5809357.

**Funding:** This work was supported by the Swiss National Science Foundation (grant FN 310030_152724/1) to SB. The funders had no role in study design, data collection and analysis, decision to publish, or preparation of the manuscript.

**Competing interests:** The authors have declared that no competing interests exist.

relevant if the SNP-wise effects from both GWAS tend to have the same sign and magnitude in the gene window. This is different from the commonly used approach asking only for the aggregate signals from two GWAS to be jointly significant. Our method is feasible due to novel insight into the product-normal distribution. We apply our new method to test for co-significant genes for severe COVID-19 and conditions leading to the prescription of common medications. Of the 23 medication classes we tested for coherent co-significant genes, only one, M05B (drugs affecting bone structure and mineralization), yielded Bonferroni significant hits. We then searched for available GWAS data for related conditions and found that also rheumatoid arthritis, calcium concentration, and vitamin D are traits pointing to several co-relevant genes in our new coherence analysis. Furthermore, testing for anti-coherence showed that the medication classes H03A (thyroid preparations), R03A, and R03BA (drugs for obstructive airway diseases) feature Bonferroni co-significant genes. Our joint analysis provides new insights into potential COVID-19 disease mechanisms.

This is a *PLOS Computational Biology* Methods paper.

## 1 Introduction

Genome-wide association studies (GWAS) correlate genotypes, most commonly single nucleotide polymorphisms (SNPs), with a phenotype of interest, both measured in the same study population. For human studies between 1 and 10 million SNPs are usually considered, and in most GWAS, each SNP is tested independently for correlation with the phenotype. By now, thousands of such GWAS have been conducted that identified a plethora of statistically significant associations of SNPs with complex traits. For most traits—in particular complex diseases or their risk factors that have been assessed in very large cohorts (100K or more subjects)—hundreds of SNPs usually turn out to be significant, even after stringent correction for multiple hypotheses testing. Individual SNP-wise effect sizes are often very small but add up to sizable narrow-sense heritability, pointing to a polygenic genetic architecture [1]. Mapping SNP-wise effects on genes and annotated gene-sets (*pathways*) [2, 3] can yield valuable insights into the genetic underpinning and potential pathomechanisms of complex diseases and aid drug discovery and repurposing.

However, due to Linkage Disequilibrium (LD) (*cf.*, [4]), many SNPs close to each other are not independent, leading to dependencies between the observed SNPs' effect sizes. This is of particular relevance when aggregating SNP-wise effects on genes or pathways. Gene-wise effects are typically computed by adding up the (squared) effects of all SNPs within the transcript region of a gene of interest, as well as sizable upstream and downstream regions that may contain regulatory elements of this gene. Pathway effects are computed from the gene-wise effects [2]. LD can lead to signal inflation for significant SNPs in sizable LD blocks since their signals are not independent. Such SNPs will dominate the joint association signal and lead to gene and pathway scores reflecting the level of importance for the phenotype inaccurately if no correction is applied. Techniques and tools have thus been developed to correct for LD in the aggregation process, such as *Pascal* [2] and *MAGMA* [3]. These tools mainly differ in their mappings of SNP effect sizes on genes, how they account for the LD structure, and details of the numerical procedure to estimate significance.

Some SNPs are significantly associated with more than one trait. The phenomenon of a single genetic variant affecting two or more traits is called *pleiotropy* [5]. In the case of disease traits, such a shared genetic component hints at the same functional pathology contributing to several diseases [6]. At the gene level, a gene is usually considered relevant for two different traits if it carries one or more effect sizes significant in both traits. However, this may not be a good criterion under all circumstances. For instance, protein-coding genes often contain several independent LD blocks. Therefore, two traits may associate with genetic variation in two or more functionally different blocks of SNPs within the same gene, which may independently be significant. Hence, even though the two traits share the same significant gene, they may not share the same genetic mechanism. To call a gene pleitropic, one should therefore move beyond comparing single variants, and take all SNPs in the gene region into account, corrected by LD.

Several methods have been proposed to uncover the shared genetic origin of two traits from GWAS summary statistics: One early method is a test of co-localization between GWAS pairs based on Bayesian statistics [7]. This method assumes that at most one association is present for each trait in the region of interest. However, the extension to the general case of multiple associations (usually the case) appears to be non-trivial. A more recent method is cross-trait *LD score regression* [8], an extension of single-trait *LD score regression* (LDSR) [9], which is a method to estimate heritability and confounding biases from GWAS summary statistics. Like single-trait LDSR, cross-trait LDSR considers the effect sizes as random variables and uses *LD-scores* (*i.e.*, the sum of genetic correlations between a given SNP and all other SNPs) to estimate the genetic correlation between two traits. Yet, these LDSR estimates typically are at the whole genome level. While restriction to genomic regions is possible, using it to obtain estimates at the individual gene level is difficult. The reason is that SNPs in the same narrow genomic region are often in high LD, so the variables entering the linear correlation may be highly dependent, something that has to be corrected for. Similarly, the standard errors and *p*-values are estimated via resampling (jackknife). This requires independent SNPs, which is not the case for strong LD. Shi *et al.* have introduced another method to estimate local genetic covariance and correlation [10]. However, this method requires the computation of the inverse SNP-SNP correlation matrix, which often does not have full rank. A regularization scheme can address this issue, but this introduces additional assumptions and parameters, which may have to be tuned for best performance. Furthermore, this method relies on decomposing the genome into independent LD blocks, usually larger than single genes.

Here, we propose the sum over the products between the effects of two traits for SNPs within a gene region as a simple measure for pleiotropy. For a single SNP, the test statistic is a simple product that is tested against the product-normal distribution, corresponding to a multiplicative meta-analysis, rather than an additive one like Fisher's. For multiple SNPs, our measure corresponds to the (non-centered) covariance between two sets of effect sizes. Importantly, we show that under the null hypothesis the corresponding test distribution can be expressed as a linear combination of $\chi^2$ distributions, with a mixture of positive and negative coefficients. This holds even if the effect sizes are not independent of each other due to LD, *i.e.*, if there exists a non-trivial covariance structure between the corresponding SNPs. The corresponding cumulative distribution function can be efficiently calculated with Davies' algorithm at high precision [11, 12]. Thus, our method considers not only isolated significant SNPs, but all SNPs within the gene region to call a gene co-significant for two traits. Furthermore, using the notion of Mendelian randomization, our statistic can be extended to test for a possible causal relationship between the two traits mediated by the tested gene.

We demonstrate the utility of our methods by a timely co-analysis of GWAS summary statistics on the severity of COVID-19, being prescribed one of 23 different medication classes, and traits related to osteoporosis, like vitamin D and calcium concentrations.

## 2 Results

### 2.1 Coherence test

Consider the index $I = \sum_i w_i z_i$, with $w_i$ and $z_i$ $N$ independent samples of two random variables $z, w \sim \mathcal{N}(0, 1)$. The index can also be written as $I = N\,\mathbb{E}(wz)$, with $\mathbb{E}$ denoting the expectation. Clearly, $I$ is proportional to the standard empirical covariance of $w$ and $z$.

For independent pairs of samples, the sampling distribution of $I$ is simply a sum of independent product-normal distributions. Hence one can infer that for identically correlated random variables with correlation coefficient $\varrho$ (*cf.*, Methods, *Product-Normal distribution*)

$$I \sim \frac{1 + \varrho}{2}[\chi_N^2] - \frac{1 - \varrho}{2}[\chi_N^2]. \tag{1}$$

In particular, for $\varrho = 0$, we have that $I \sim VG(N, 1)$, with $VG$ being the *variance-gamma* distribution discussed in more detail in S1 Text. One should note that the difference is in the distributional sense and, therefore, generally non-vanishing. A null hypothesis of zero correlation (or some other fixed value) can therefore be tested, as the cumulative distribution function (cdf) for $I$ can be calculated explicitly and efficiently with Davies' algorithm (*cf.*, Methods, *Linear combination of $\chi^2$ distributions*).

The main advantage of the above significance test is that it is straightforward to relax the requirement of sample independence. That is, we can view the index $I$ as a scalar product of random samples of $w \sim \mathcal{N}(0, \Sigma_w)$ and $z \sim \mathcal{N}(0, \Sigma_z)$, with $\mathcal{N}$ denoting here the multivariate Gaussian distribution and $\Sigma_{w|z}$ covariance matrices. For $\Sigma_w = \Sigma_z = 1$, the corresponding distribution of $I$ is given by (1). In the general case, the inter-dependencies can be corrected via linear decorrelation, *cf.*, Methods *Coherence test decorrelation*. The case of interest for this paper is $\Sigma = \Sigma_w = \Sigma_z$. In this case, one can show that under the null of $w$ and $z$ being independent

$$I \sim \sum_i \frac{\lambda_i}{2}[\chi_1^2] - \sum_i \frac{\lambda_i}{2}[\chi_1^2], \tag{2}$$

with $\lambda_i$ the $i$th eigenvalue of $\Sigma$. Hence, $I$ is distributed according to a linear combination of $\chi_1^2$ distributions with positive and negative coefficients, and therefore the cdf and tail probability can be calculated with Davies' algorithm. Note that the above discussion can be extended to non-standardized variables (for $\varrho = 0$) via the results given in section 1 in S1 Text.

As discussed in detail in [2], a GWAS (*cf.*, Methods, *GWAS*) gene enrichment test can be performed via testing against $I$ with $w = z$. This effectively tests against the expected variance of SNPs' significances in the gene. Here we propose to use $I = w^T z$ with $w$ and $z$ resulting from two different GWAS phenotypes to test for the co-significance of a gene for two GWAS. A significance test can be performed either against the right tail of the null distribution (2) (*coherence*) or against the left tail (*anti-coherence*). For a single SNP in the gene window, the test reduces to the *product-normal* test, whose properties will be discussed in more detail in the following section 2.2.

Note that we do not centralize $w$ and $z$ over the gene SNPs. Hence, we do not test for covariance but for a non-vanishing second cross-moment. After decorrelation, it is best to interpret this as testing each joint SNP within the gene independently for a coherent deviation from the null hypothesis. Therefore we refer to this test as testing for genetic coherence or simply as cross-scoring. For simplicity, we only consider a fixed effect size model and assume that the correlation matrix $\Sigma$ obtained from an external reference panel is a good approximation for both GWAS populations.

The above coherence test assumes that there is no sample overlap between the populations of the pair of GWAS considered. However, the presence of sample overlap can be corrected for, as discussed in section 4 in S1 Text.

## 2.2 Simulation study

It is useful to discuss the single SNP case in more detail. The distribution (2) simplifies for $N = 1$ to $I \sim \frac{1}{2}[\chi_1^2] - \frac{1}{2}[\chi_1^2]$, which corresponds to the uncorrelated product normal distribution, *cf.*, Methods Eq (8). The index $I$ for $N = 1$ is a measure of coherence (or anti-coherence) between $z$ and $w$. The significance threshold curve for a fixed desired $p$-value, say $p_I = 10^{-7}$, is illustrated in Fig 1. The curve corresponding to a given $p_I$ is unbounded. For a given $w$, there is always a corresponding $z$ such that the resulting product $I$ is significant. This differs from Fisher's exact method which combines two $p$-values $p_{w, z}$ into a combined one ($p_F$) via $-2 \log p_F = -2 \log p_w - 2 \log p_z \sim [\chi_4^2]$. Since Fisher's method combines significance by addition, the corresponding combined significance curve is bounded. Specifically, in the extreme case of one of the $p$-values being equal to one, the significance threshold is finite and fixed by the other $p$-value. In contrast, in the case of the product-normal, the divergence between the $p$-values is penalized. If one of the $p$-values is large, say $p_w \simeq 1$, the other one has to be extremely small to achieve a given combined significance value, i.e. $p_z \ll p_w$ (*cf.*, Fig 1). Therefore, the product-normal-based method becomes more and more conservative for increasingly diverging $p$-values. In contrast, for similar $p$-values, our method is less restrictive than Fisher's, and for identical $p$-values its effective threshold is almost one order of magnitude lower (*cf.*, Fig 1). Hence, one should see Fisher's method as additive in the evidence, while the product-normal-based method is multiplicative.

The importance of correcting for the inter-dependence between $w$ and $z$ elements in the index $I$ is already visible for $N = 2$, as discussed in section 3 in S1 Text. For higher $N$, this becomes even more pronounced. For example, consider a correlation matrix $\Sigma$ of dimension one hundred with off-diagonal elements identically set to 0.2. We draw 1000 pairs of

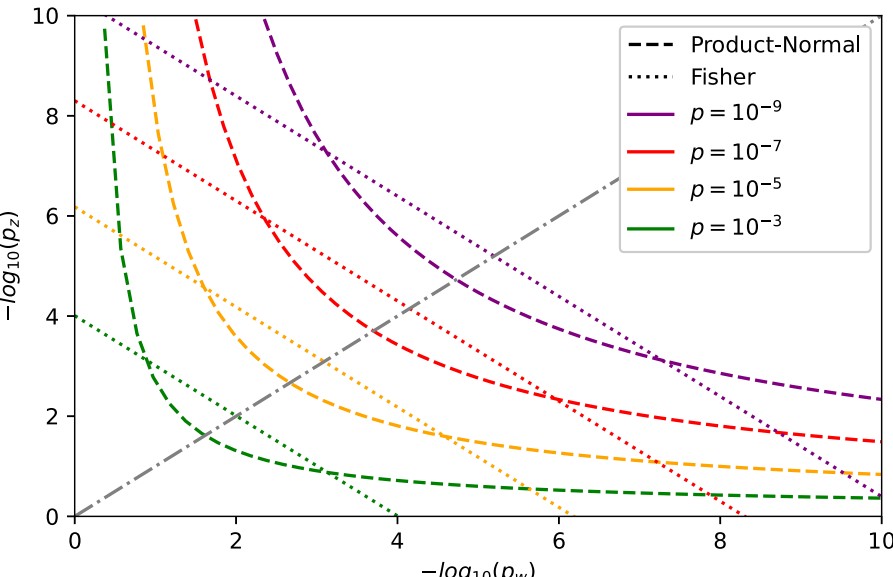

**Fig 1. Significance threshold curves in the one element case for the product-normal (dashed) and Fisher's method (dotted) for various *p*-values.** The diagonal is indicated by a gray dash-dotted line and corresponds to equal $-\log_{10}$ *p*-values.

independent samples of $\mathcal{N}(0, \Sigma)$ and calculate $I$ for each pair. A $p$-value is then obtained for each index value for the linear combination of $\chi^2$ distributions (2) (also referred to as weighted $\chi^2$), and for the variance-gamma distribution. Recall that the latter does not correct for the off-diagonal correlations. We repeat the experiment with the off-diagonal elements set to 0.8, resulting in a stronger element-wise correlation. The resulting qq-plots for both cases are shown in 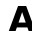 Fig 2.

**A**

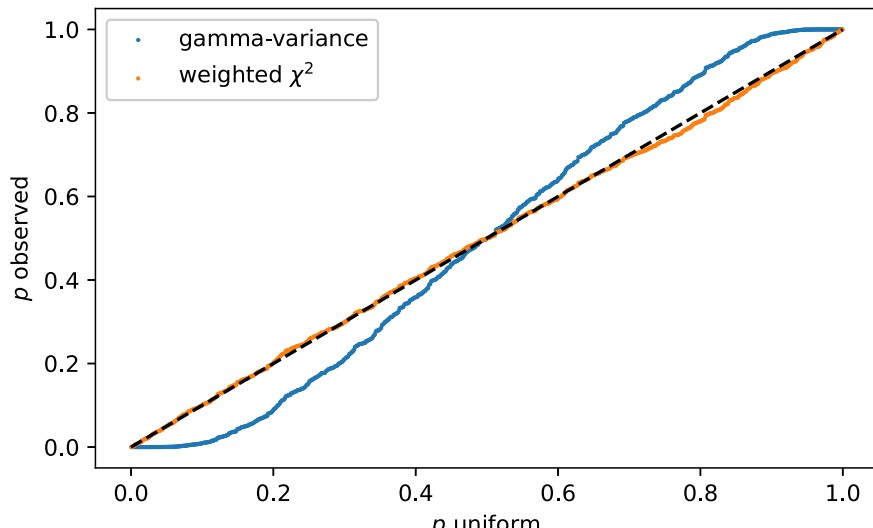

**B**

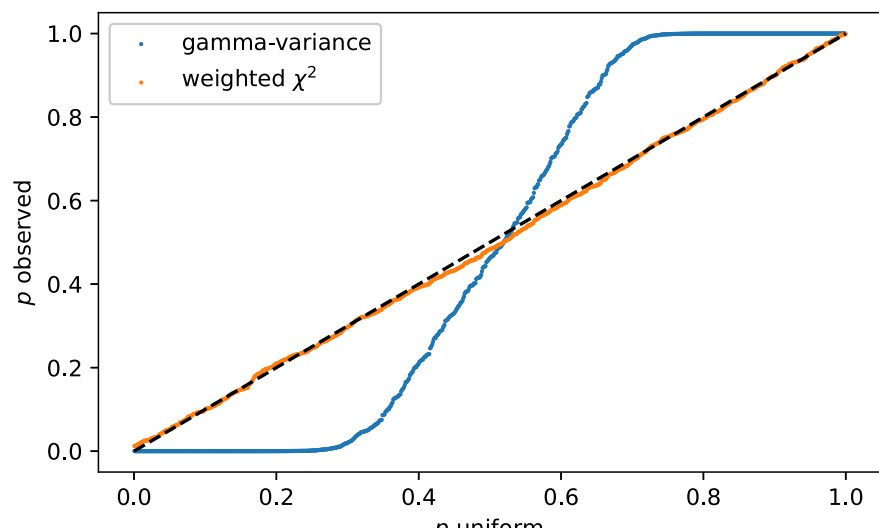

**Fig 2. QQ-plots of observed $p$-values resulting from the index $I$ for 1000 pairs of samples of $\mathcal{N}(0, \Sigma)$ against uniform $p$-values.** A: Off-diagonal elements of $\Sigma$ set to 0.2. B: Off-diagonal set to 0.8. The blue curve is obtained using the variance-gamma distribution to perform the statistical test, while the orange curve is obtained via the weighted $\chi^2$ distribution. The latter corrects for the correlation and therefore is well calibrated.

We observe that the variance-gamma distribution (1) (with $\varrho = 0$) indeed becomes unsuitable for increasing element-wise correlation of the data sample elements. Not correcting for the inter-sample correlation leads to more and more false positives with increasing correlation strength. In contrast, the weighted $\chi^2$ distribution (2) yields stable results in both the weakly and strongly correlated regime, as is evident in Fig 2.

### 2.3 Ratio test

Consider the normalized index

$$R = \frac{\sum_i w_i z_i}{\sum_j z_j^2} \; , \tag{3}$$

with $w$ and $z$ as in the previous sections. In particular, being independent random variables. The cdf for $R$ can be calculated to be given by

$$F_R(r) = \Pr(R \leq r) = F_{\hat{v}\hat{z}}(0), \tag{4}$$

with $\hat{z} \sim \mathcal{N}(0, \Lambda)$, $\hat{v} \sim \mathcal{N}(0, (1 + r^2)\Lambda)$, $\Lambda$ the matrix of eigenvalues of $\Sigma$, and $F_{\hat{v}\hat{z}}(0)$ the linear combination of $\chi_1^2$ cdf evaluated at the origin, see Methods, *Ratio test derivation* (Similar results can be obtained for $w$ and $z$ interchanged). Hence, the cdf of the ratio (3) can also be calculated with Davies' algorithm. A consistency check follows from the case of one dimension, where $R$ has to be Cauchy distributed. The corresponding cdf is given by $F_C = 0.5 + \arctan(r)/\pi$. Evaluation for various $r$ shows agreement with values calculated from (14) in the one-dimensional case. Note that similar expressions can be derived for non-standardized $w_i$ and $z_i$, albeit in terms of the non-central $\chi^2$ distribution, *cf.*, section 1 in S1 Text.

Note that the ratio $R = w^T z/(z^T z) = \sum_i \lambda_i \bar{w}_i \bar{z}_i/(\sum_j \lambda_j \bar{z}_j^2)$ with $\bar{w}_i$ and $\bar{z}_i$ i.i.d. $\mathcal{N}(0, 1)$, and $\lambda_i$ the $i$th eigenvalue of $\Sigma$, can be interpreted as the weighted least squares solution in the case of heteroscedasticity for the regression coefficient of the linear regression between the de-correlated effect sizes of the two GWAS. Therefore, with the cdf for $R$ derived above, we can test for a significant deviation from the null expectation of no relation. In general $R$ is not invariant when swapping $w$ and $z$ and may be used under certain conditions to make inference about the causal direction, *cf.*, multi-instrument Mendelian randomization, in particular [13]. Specifically, we consider the trait related to $w$ as exposure and that of $z$ as the outcome. Then the ratio test tries to detect genes that are strongly (anti-)coherent between exposure and outcome but at the same time only exhibit relatively low variance in the outcome, as is clear from the definition of $R$. Thus the ratio test effectively normalizes the alignment of the outcome to the exposure, with respect to the outcome's own variation. The causal direction from exposure to outcome is implied if the exposure is confirmed to be associated with the gene via a significant gene enrichment $p$-value, $p_V$, obtainable from the usual $\chi^2$ test of [2], and if confounding factors can be excluded. The overall scheme is illustrated in Fig 3.

Similar to the previously introduced coherence test, we assumed above that the populations of the pair of GWAS considered present no overlap. However, sample overlap can be corrected for as well, *cf.*, section 4 in S1 Text.

### 2.4 COVID-19 GWAS application

To demonstrate the usefulness of our methods in the context of actual and topical GWAS data (*cf.*, Methods, *GWAS*), we considered the recent meta-GWAS on very severe respiratory confirmed COVID-19 [14, 15] as the primary phenotype and co-analyzed it with several other related traits (see also *Data availability* statement). Specifically, we used the summary statistics

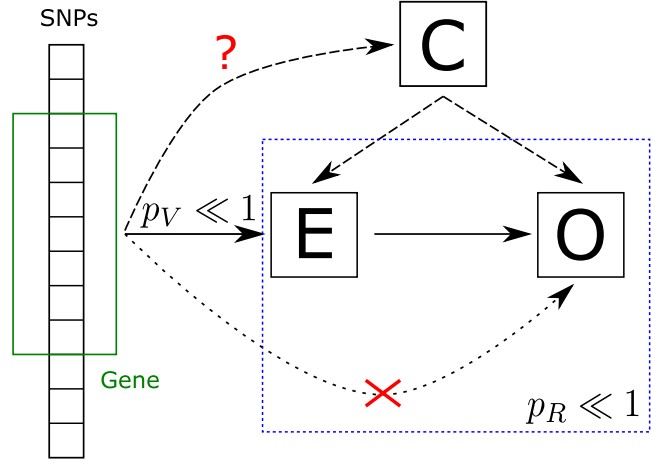

**Fig 3. Interpretation of the ratio test.** The test detects potential gene-wise causal relations between a GWAS trait viewed as exposure (E) and a GWAS trait viewed as outcome (O). The association of the exposure has to be confirmed independently via a standard gene enrichment test. Potential confounders (C) have to be excluded by other means.

resulting from European subjects, excluding those from UK Biobank (A2_ALL_eur_leave_ukbb_23andme) in order to avoid overlap with the secondary trait GWAS. This GWAS shows significant gene enrichment on chromosomes 3 and 12, as can be inferred via testing for gene enrichment following [2]. The Manhattan plot for the resulting gene $p$-values is shown in S1 Fig.

We cross-scored this GWAS against a panel of GWAS on medication within the UK Biobank [16], which take prescription (self-reported intake) of 23 common types of medications as traits. Hence, in cross-scoring against the severe COVID-19 GWAS, we sought to uncover whether there is a shared genetic architecture of predisposition for severe COVID and being prescribed specific medications. The description of all medication class codes used can be found in S1 Table. Note that, as usual, these GWAS have been performed with age as one of the covariates. Therefore, in our pair-wise analysis age-dependent effects are already regressed out and do not play a significant role in our analysis.

All calculations were performed with the python package *PascalX* [17], which incorporates the methods detailed in the *Methods* section (see also *Code availability* statement). The association directions were extracted from the signs of the raw effect sizes ($\beta$). As a reference panel, we used the European sub-population of the high coverage release of the 1K Genome project [18]. Note that we took only SNPs with matching alleles between the GWAS pairs and the reference panel into account. We transformed the raw GWAS $p$-values via *joint* rank transformation, for the reason discussed in Methods under *SNP normalisation*. We tested all genes with at least one SNP present in both traits, and defined the gene region as the transcription side plus a window of 50kb on both sides in order to capture as well potential regulatory effects. We verified at hand of one drug class (M05B) that the choice of gene window does not lead to an inflation of signal, see S2 Fig.

The cross-scoring results for the test are shown in Fig 4.

Cross scoring revealed joint coherent signals between GWAS effects from the prescription of group M05B medications (drugs affecting bone structure and mineralization) and that for very severe COVID-19. We show the Manhattan plot resulting from the null model (2) for the medication group M05B in Fig 5 (for the corresponding qq-plot, see S3 Fig).

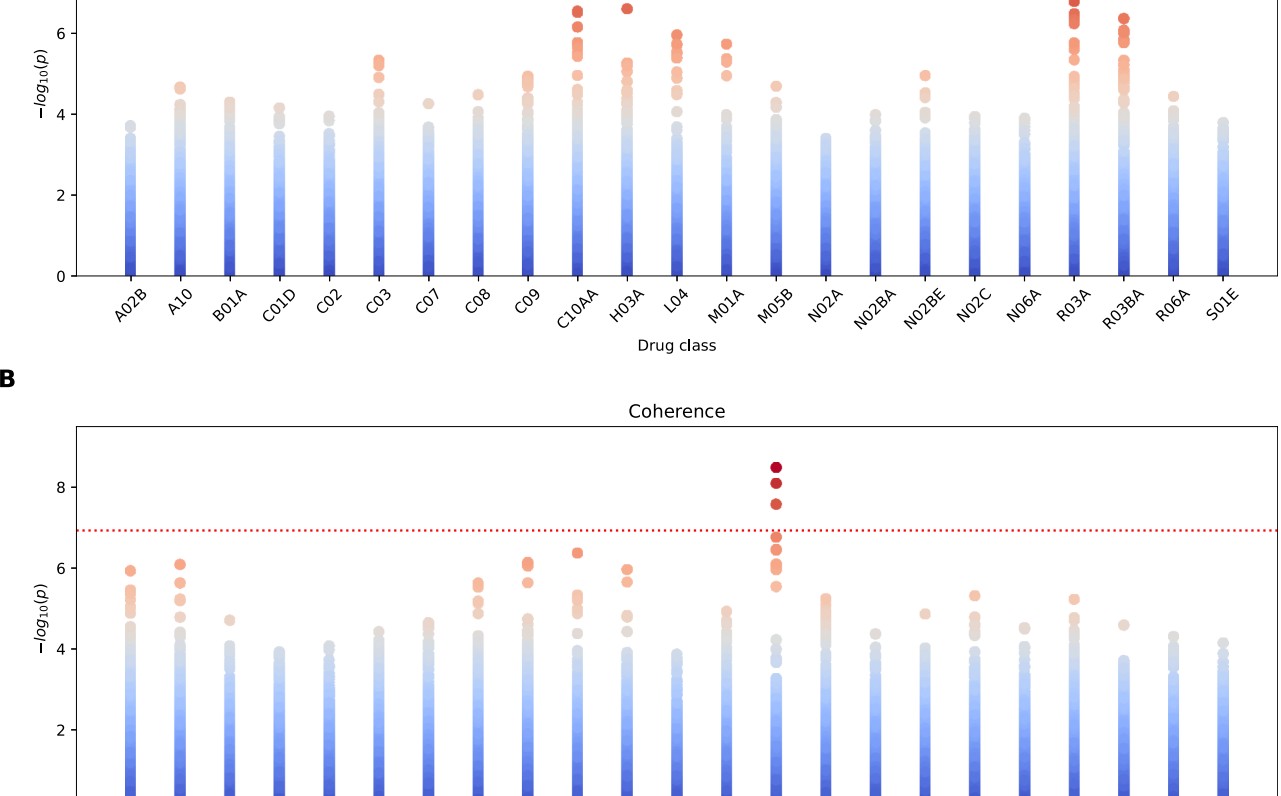

**Fig 4. Resulting *p*-values for cross-scoring 23 drug classes GWAS against very severe COVID-19 GWAS for anti-coherence (A) and coherence (B).** Each data point corresponds to a gene. The dotted red line marks a Bonferroni significance threshold of $1.18 \times 10^{-7}$ (0.05 divided by the 18453 genes tested and 23 drug classes). Note that the drug class M05B shows the most significant enrichment in coherence with severe COVID-19.

We observe that SNP-wise effects in genes in the well-known COVID-19 peak locus on chromosome 3 appear to be coherent with those from being prescribed M05B medication, with Bonferroni significance for the chemokine receptor genes *CCR1*, *CCR3* and the gene *LZTFL1* in the region chr3p21 (we Bonferroni corrected for number of genes and drug classes tested). We tested the orientation of the aggregated associations of these genes via the D-test, *cf.*, Methods, *Direction of association*, over the COVID-19 GWAS and find a positive direction (right tail) with $p_D \simeq 1.7 \times 10^{-4}$, $8.6 \times 10^{-3}$ and 0.03, respectively.

For illustration, we show the spectrum of SNPs considered in the *CCR3* region and their SNP-SNP correlation matrix in Fig 6. One can see a large block of SNPs in high LD, encompassing some of *CCR3*'s 5'UTR and most of its gene body, all having positive associations with both severe COVID-19 and M05B drug prescription. Furthermore, a somewhat weaker coherence signal can be seen for SNPs in its 3'UTR, some of which share negative associations.

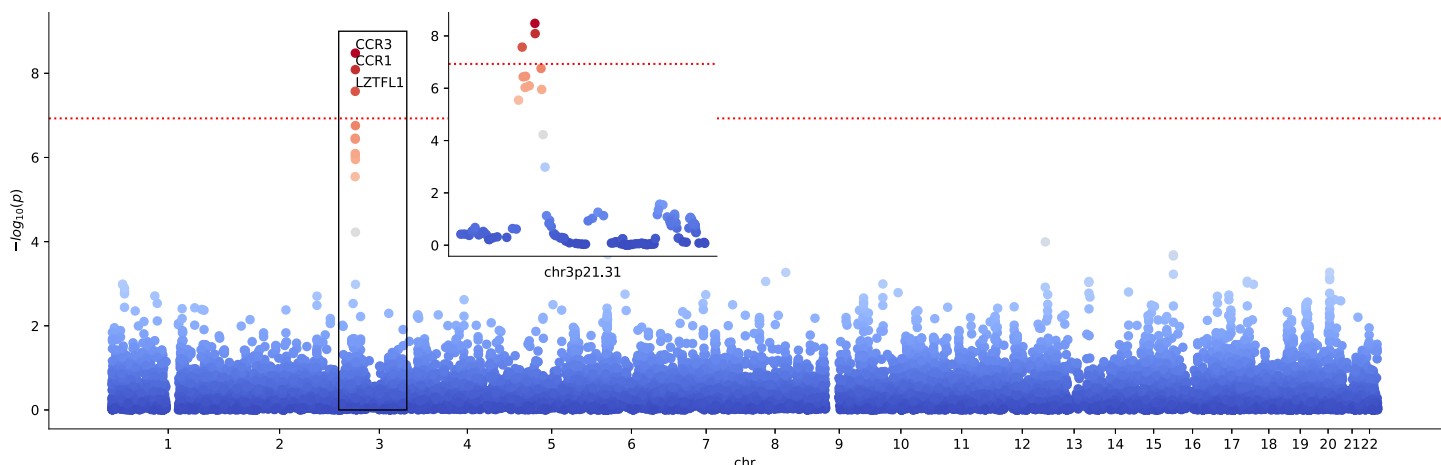

**Fig 5. Manhattan plot for cross scoring very severe confirmed COVID-19 with medication class M05B for coherence.** Data points correspond to genes. The dotted red line marks a Bonferroni significance threshold of $1.18 \times 10^{-7}$ (0.05 divided by the 18453 genes tested and 23 drug classes). The inlay plot shows a zoom into the relevant locus chr3p21.31.

Using our coherence test described above, we calculate a *p*-value of $p \simeq 8.2 \times 10^{-9}$ for *CCR1* and $p \simeq 3.3 \times 10^{-9}$ for *CCR3*.

The SNP spectrum in the *LZTFL1* region is shown in S4 Fig. The *p*-value for the significance of the coherence is $p \simeq 2.7 \times 10^{-8}$.

The directions of associations inferred from the D-test and their coherence suggest that genetic predispositions leading to M05B prescription may carry a higher risk for severe COVID-19. We list the Bonferroni significant pathways for (anti)-coherent M05B and severe COVID-19 genes detected via a pathway enrichment test (Methods, *Pathway enrichment*) in

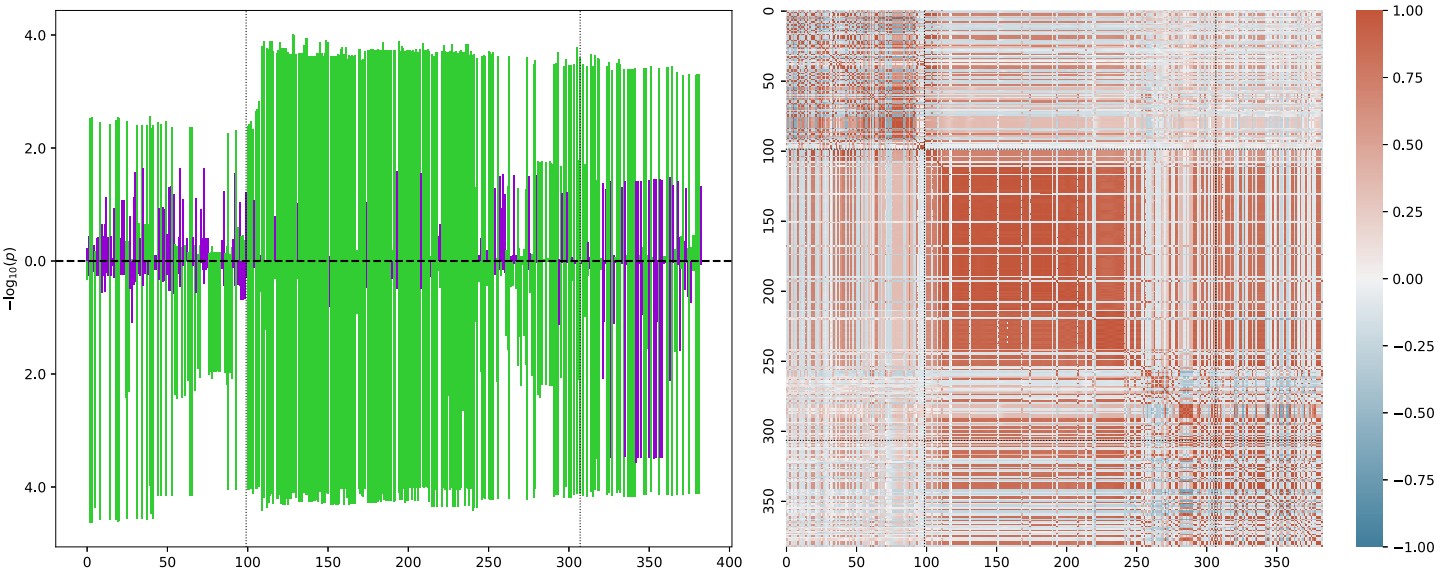

**Fig 6. Left: SNP *p*-values after rank transform for the *CCR3* gene.** The *x*-axis is numbered according to the *i*th SNP in the gene window ordered by increasing position. The dotted black lines indicate the transcription start and end positions (first and last SNP). Up bars correspond to M05B and down bars to severe COVID-19. Green indicates positive and violet negative association with the trait. Right: SNP-SNP correlation matrix inferred from the 1KG reference panel. The color scale is shown on the right with dark red (+1) corresponding to perfect SNP-SNP correlation and dark blue (−1) to perfect anti-correlation over the reference population.

Table 1. Note that the enrichment test of pathways is performed with the (anti)-coherent gene enrichment $p$-values, and therefore tests for pathways that are relevant for both traits simultaneously (we Bonferroni corrected for number of pathways and number of traits we tested for pathway enrichment)

The cross-scoring test also detected the gene *HLA-DQA1* to be Bonferroni significant under anti-coherence for the drug classes H03A (thyroid preparations), as well as R03A and R03BA (drugs for obstructive airway diseases). Further, we note that the drug classes C10AA (HMG CoA reductase inhibitors) and L04 (immunosuppressants) have gene hits with a $p$-value $<1 \times 10^{-5}$. Interestingly, all these drug classes possess indications in autoimmune diseases and allergies. In particular, of the non-significant genes we detect the gene *FUT1* with $p \simeq 2.9 \times 10^{-7}$ for C10AA to be closest to Bonferroni significance. The D-test shows that *FUT1* is located in the left-tail under C10AA ($p_D \simeq 9.8 \times 10^{-4}$), and therefore this gene's variations may be a possible risk factor for severe COVID-19. It is known that *FUT1* is involved in creating a precursor of the H antigen, itself a precursor to each of the ABO blood group antigens. It has been hypothesized before in the literature that the ABO blood system correlates to COVID-19 severity [19].

We also find the observed leading *TRIM* genes for L04 of interest. For illustration of the anti-coherence case, the SNP spectrum for *TRIM26* ($p \simeq 1.1 \times 10^{-6}$) under L04 is shown in S5 Fig. The pathway enrichment test in the anti-coherent case shows that L04 cross-scored severe COVID-19 has Bonferroni significant enrichment in interferon gamma-related pathways, see Table 1. Note that *TRIM* proteins are expressed in response to interferons [20], and therefore the detected pathways are consistent with the enrichment of *TRIM* genes.

## 2.5 Osteoporosis related GWAS

The main application of M05B medications is the treatment of osteoporosis. In order to investigate this potential link further, we cross-scored the COVID-19 GWAS against a selection of GWAS with phenotypes related to osteoporosis, namely, bone mineral density (BMD) estimated from quantitative heel ultrasounds and fractures [21], estrogen levels in men (estradiol and estrone) [22], calcium concentration [23], vitamin D (25OHD) concentration [24] and rheumatoid arthritis (RA) [25]. The inferred gene enrichments for coherence and anti-coherence with the COVID-19 GWAS are shown in Fig 7.

We found Bonferroni significant enrichment for RA and enrichment with gene $p$-values $< 10^{-5}$ for calcium and vitamin D (we Bonferroni corrected for number of genes and traits tested). The Manhattan plot for the Bonferroni significant trait is shown in Fig 8 (for the qq-plot, see S6 Fig). Note that $\approx 47\%$ of the SNP alleles between the RA and COVID-19 GWAS were not matching and that we discarded all non-matching SNPs for the analysis.

A list of the most significant detected genes under the test is given in Table 2. The table also contains a brief description of the potential relation of the respective gene to COVID-19, if known.

For RA, several TRIM genes are Bonferroni significant. TRIM proteins are associated with innate immunity, and are in particular involved in pathogen recognition and host defense [20, 26]. This is consistent with the (weaker) TRIM signals we detected for the immunosuppressants L04, which are indicated in the treatment of RA. The D-test shows that the leading TRIM genes are in the right tail under the RA GWAS, with $p_D < 10^{-6}$. Therefore, the anti-coherence suggests a protective effect of variants related to a predisposition for RA.

For the calcium anti-coherent case, the top gene *HGFAC* with $p \simeq 4.0 \times 10^{-7}$ is only slightly below the Bonferroni significance threshold. This gene sits in the right tail ($p_D \simeq 8.9 \times 10^{-4}$) under the D-test (12) applied to the calcium GWAS. Therefore, the aggregated variants in this

**Table 1. Bonferroni significant pathways for cross-scored severe COVID-19 genes.** We only tested the six listed traits for pathway enrichment as these traits possess significant or close to significant genes under the coherence test. The coherent case corresponds to the right (R) tail, the anti-coherent case to the left (L) tail. The number of genes in the pathway is given in the fourth column. We tested against the 32284 gene sets of MSigDB 7.4. We list Bonferroni significant ($p < 0.05/32284/6 \simeq 2.6 \times 10^{-7}$) pathways and close to significant pathways.

| trait | tail | enriched pathways | # genes | $p$-value |
|---|---|---|---|---|
| C10AA | R | - | | |
| | L | - | | |
| H03A | R | - | | |
| | L | KEGG graft versus host disease | 21 | $1.1 \times 10^{-8}$ |
| | | KEGG autoimmune thyroid disease | 25 | $3.4 \times 10^{-8}$ |
| | | GOCC MHC protein complex | 9 | $3.4 \times 10^{-8}$ |
| | | KEGG allograft rejection | 22 | $4.1 \times 10^{-8}$ |
| | | GOBP interferon gamma mediated signaling pathway | 72 | $5.8 \times 10^{-8}$ |
| | | GOMF MHC class II receptor activity | 3 | $6.5 \times 10^{-8}$ |
| | | GOCC lumenal side of endoplasmic reticulum membrane | 16 | $7.7 \times 10^{-8}$ |
| | | Durante adult olfactory neuroephithelium B cells | 30 | $1.0 \times 10^{-7}$ |
| | | HP hemoptysis | 54 | $2.9 \times 10^{-7}$ |
| | | GOCC MHC class II protein complex | 3 | $3.0 \times 10^{-7}$ |
| | | GOCC lumenal side of membrane | 23 | $3.0 \times 10^{-7}$ |
| | | GOMF peptide antigen binding | 14 | $3.7 \times 10^{-7}$ |
| | | HP abnormal pulmopnary thoracic image finding | 66 | $4.9 \times 10^{-7}$ |
| | | GOBP response to interferon gamma | 148 | $8.2 \times 10^{-7}$ |
| | | GOBP regulation of leukocyte proliferation | 214 | $8.6 \times 10^{-7}$ |
| | | KEGG type I diabetes mellitus | 27 | $1.5 \times 10^{-6}$ |
| L04 | R | Farmer breast cancer cluster 8 | 3 | $1.2 \times 10^{-6}$ |
| | L | GOBP interferon gamma mediated signaling pathway | 72 | $1.5 \times 10^{-9}$ |
| | | GOMF peptide antigen binding | 14 | $1.3 \times 10^{-7}$ |
| | | Reactome interferon gamma signaling | 67 | $2.0 \times 10^{-7}$ |
| | | GOBP response to interferon gamma | 148 | $4.8 \times 10^{-7}$ |
| M05B | R | Roeth tert targets dn | 8 | $7.3 \times 10^{-8}$ |
| | L | - | | |
| R03A | R | - | | |
| | L | GOBP antigen processing and presentation of endogenous peptide antigen | 14 | $1.5 \times 10^{-7}$ |
| | | GOBP antigen processing and presentation of endogenous antigen | 17 | $3.4 \times 10^{-7}$ |
| R03BA | R | - | | |
| | L | KEGG graft versus host disease | 21 | $6.4 \times 10^{-9}$ |
| | | Module 143 | 9 | $1.4 \times 10^{-7}$ |
| | | GOCC MHC protein complex | 9 | $3.8 \times 10^{-7}$ |
| | | Module 293 | 7 | $5.3 \times 10^{-7}$ |
| | | WP cytokines and inflammatory response | 22 | $6.5 \times 10^{-7}$ |
| | | KEGG allograft rejection | 22 | $7.6 \times 10^{-7}$ |
| | | GOMF MHC class II receptor activity | 3 | $1.4 \times 10^{-6}$ |
| | | GOBP antigen processing and presentation of endogenous peptide antigen | 14 | $1.5 \times 10^{-7}$ |
| | | GOBP antigen processing and presentation of endogenous antigen | 17 | $2.3 \times 10^{-7}$ |
| | | Gaurnier PSDM4 targets | 44 | $1.1 \times 10^{-6}$ |
| | | KEGG type I diabetes mellitus | 27 | $1.6 \times 10^{-6}$ |

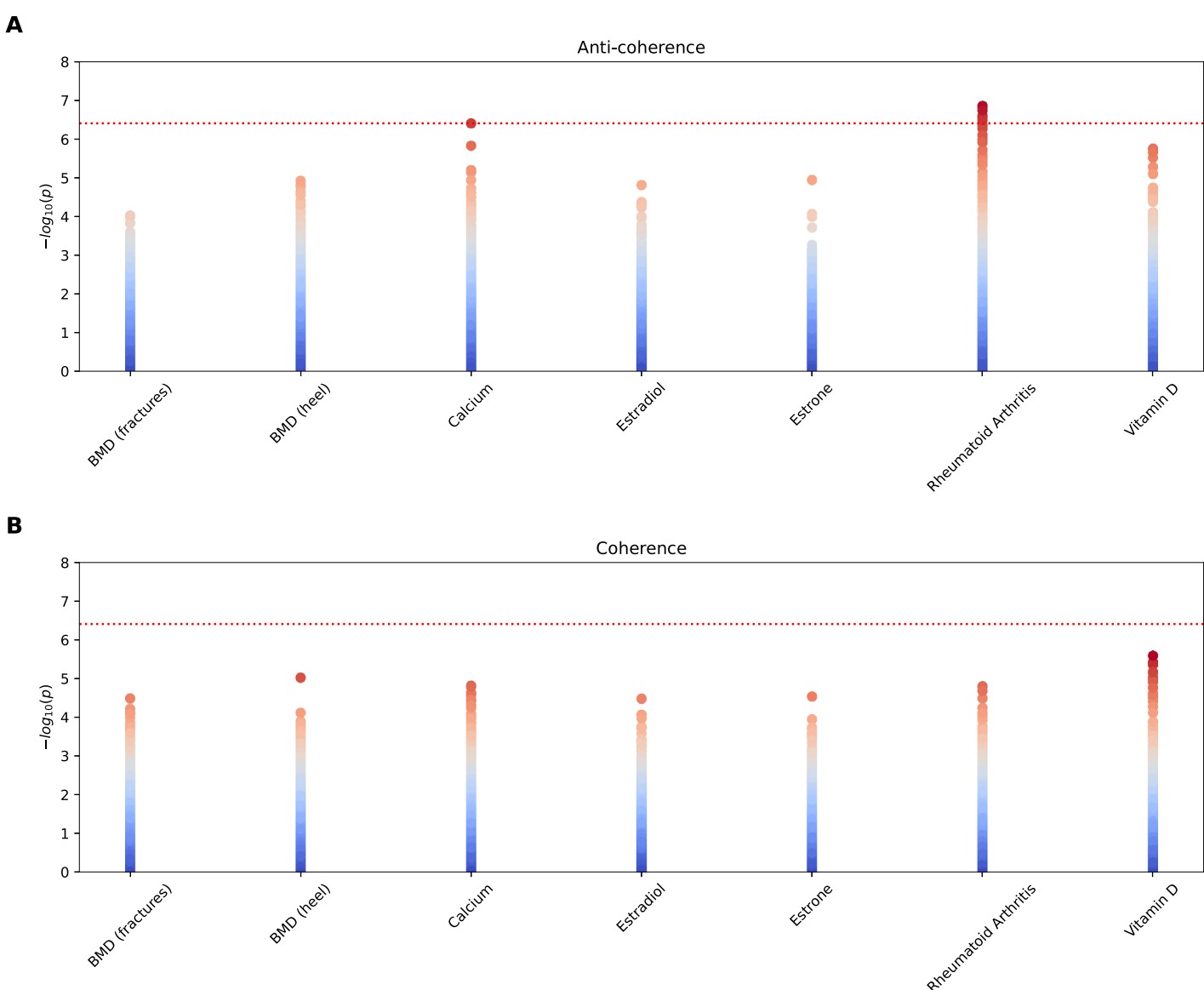

**Fig 7. Resulting *p*-values for cross-scoring several GWAS related to osteoporosis against the severe COVID-19 GWAS.** The top figure (A) shows the anti-coherent case and the bottom figure (B) the coherent case. The red dotted line marks the Bonferroni significance threshold of $3.9 \times 10^{-7}$ (0.05 divided by the 18453 genes tested and 7 traits). We observe Bonferroni significant enrichment for RA plus enrichments in vitamin D and calcium with *p*-values $< 10^{-5}$.

gene imply that a predisposition for high calcium concentration may implicate a reduced risk for severe COVID-19, *i.e.*, a protective effect.

Let us also briefly discuss an application of the ratio-based causality test. We ratio tested the COVID-19 GWAS against M05B, RA, vitamin D, and calcium in the coherent andanti-coherent case, using Eq (4). The resulting gene hits of interest are summarized in Table 3, with a brief description of the potential COVID-19 context of the gene hits. Unless not indicated otherwise in the table, confounding between calcium and vitamin D could be excluded.

The Manhattan plot resulting from the ratio test for Vitamin D as exposure and COVID-19 as outcome is shown in Fig 9 (the corresponding qq-plot is shown in S7 Fig).

We found that Vitamin D carries two interesting hits suggesting causal pathways from Vitamin D concentration to the severity of COVID-19, namely, the genes *KLC1* and *ZFYVE21*.

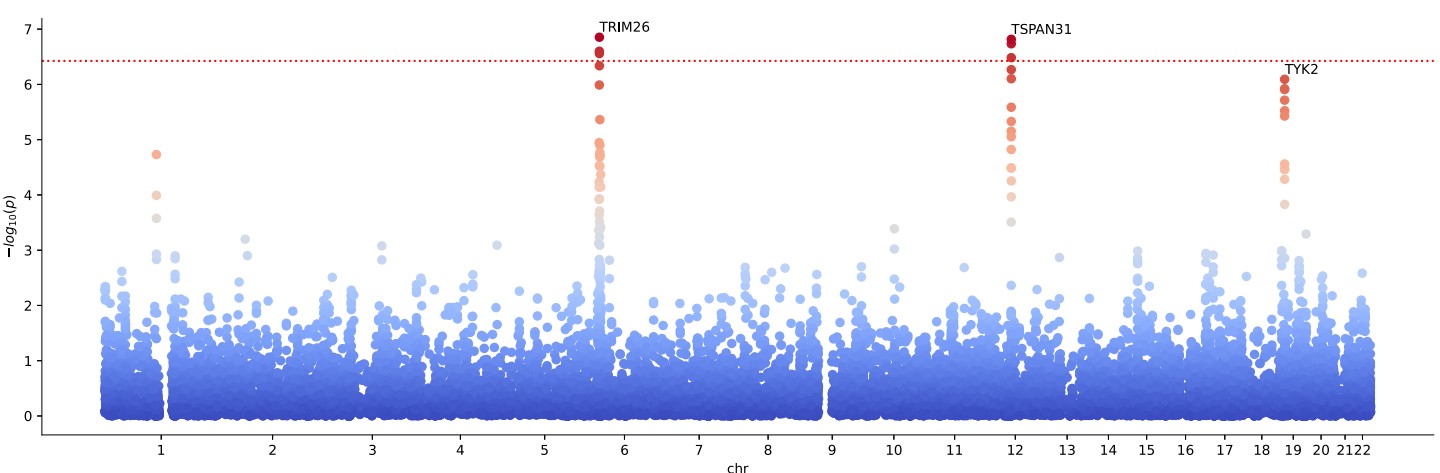

**Fig 8. Manhattan plot for cross-scoring severe confirmed COVID-19 with rheumatoid arthritis for anti-coherence.** Data points correspond to genes. The dotted red line marks a Bonferroni significance threshold of $3.9 \times 10^{-7}$ (0.05 divided by the 18453 genes tested and 7 traits). The labels denote the leading gene hit for the corresponding peak.

The observed $p_R$-values suggest a causal flow from a genetic predisposition for vitamin D concentration to the severity of COVID-19, mediated via these genes.

In the anti-coherent case, the gene *HOXC4* is detected for M05B. Modulo potential confounders, a causal relation from a predisposition to take M05B to the severity of COVID-19 via *HOXC4* is suggested. Interestingly, we also find an inverse causal relation, namely, a predisposition for severe COVID-19 implying a predisposition for calciumconcentration, mediated by the gene *DPP9*. For a brief discussion of the potential role played by these genes in the COVID-19 context, we refer toTable 3.

We also tested all cases against pathway enrichment (see Methods, *Pathway enrichment*). We only detected Bonferroni significant pathways for RA, see Table 4.

Note that while we did not detect genes of significance at the gene level under the ratio test for RA, the non-significant genes combine to Bonferroni significant pathways for the trait (we Bonferroni corrected for number of pathways and number of traits pathway enrichment has been tested for). We observed the detected pathways already in Table 1 to be co-enriched for immune system-related drug classes and severe COVID-19. The ratio test implies that these pathways may play a causal role.

## 2.6 Multiplicative meta-analysis

We investigated how the gene-wise coherence test based on the product-normal statistic compares to results from the more simple procedure of first computing gene-wise significance for each trait (via the usual $\chi^2$-test, *cf.*, [2]), and then testing the corresponding combined scores via the product-normal as if there was only a *single* genetic element affecting both traits (*cf.*, Fig 1 for an illustration of the corresponding combined product normal *p*-values). However, in doing so, we loose the information on the direction of coherence. As before, we used jointly qq-normalized GWAS *p*-values to avoid the risk of unintended uplift and only kept SNPs with consistent alleles between the GWAS pairs.

In general, we expect such a simple meta-analysis approach to generate both more false positives and false negatives. It is clear that ignoring the coherence of the contributing SNP-wise effects in the gene window can create false positives. However, false negatives may also occur: In our proposed novel coherence test, introduced in the Results section, several independent

**Table 2. Table of genes detected via the coherence test for COVID-19 against RA, vitamin D, and calcium (with $p < 6.0 \times 10^{-6}$).** The column **D** indicates the direction of the test with + for coherent and − for anti-coherent.

| gene | trait | D | p-value | COVID-19 context description |
|---|---|---|---|---|
| TRIM26 | RA | − | $1.4 \times 10^{-7}$ | TRIM proteins are associated with innate immunity and are in particular involved in pathogen recognition and host defense [20, 26]. |
| TRIM10 | | | $2.5 \times 10^{-7}$ | |
| TRIM15 | | | $2.8 \times 10^{-7}$ | |
| TRIM40 | | | $4.6 \times 10^{-7}$ | |
| TSPAN31 | | | $1.5 \times 10^{-7}$ | Has been observed before to be regulated in Vero E6 cells over-expressing the SARS-CoV S2 subunit [54]. |
| AGAP2 | | | $1.8 \times 10^{-7}$ | Modulates the transforming growth factor beta-1 (*TGF-β1*), the principal mediator of the fibrotic response in liver, lung, and kidney [55]. |
| CDK4 | | | $3.3 \times 10^{-7}$ | Cyclin-dependent kinases (CDKs) have been proposed as a new treatment option for COVID-19 [56]. |
| OS9 | | | $5.4 \times 10^{-7}$ | Codesfor a protein that binds to the hypoxia-inducible factor 1 (*HIF-1*), a key regulator of the hypoxic response [57, 58]. Regulation of *HIF-1* interpolates between regeneration and scaring of injured tissue [59]. We know that severe COVID-19 may lead to lung tissue fibrosis [60, 61]. |
| TYK2 | | | $8.1 \times 10^{-7}$ | Component of type I and type III interferon signaling pathways. The gene has been implicated before to be involved in the genetic mechanisms for critical illness due to COVID-19 [62]. |
| OAS3 | vitamin D | | $1.8 \times 10^{-6}$ | The *OAS* family are essential proteins involved in the innate immune response to viral infection [63]. Vitamin D can increase the expression of the *OAS* genes [64]. Under the D-test we have $p_D \simeq 7.2 \times 10^{-5}$ for the COVID-19 GWAS. |
| OAS2 | | | $2.2 \times 10^{-6}$ | |
| OAS1 | | | $3.0 \times 10^{-6}$ | |
| FYCO1 | | | $5.3 \times 10^{-6}$ | Plays a role in microtubule plus end-directed transport of autophagic vesicles [65]. SARS-CoV-2 inhibits autophagy activity [66]. A regulatory role of vitamin D on autophagy at different steps, including induction, nucleation, and degradation, has been suggested [67]. |
| LZTFL1 | | | $7.9 \times 10^{-6}$ | Modulates T-cell activation and enhances IL-5 production [68]. Mouse models suggest that expression of IL-5 alters bone metabolism [69]. |
| CXCR6 | | | $8.1 \times 10^{-6}$ | This gene is expressed by subsets of $T_H1$ cells, but not by $T_H2$ cells, and may be important in the trafficking of effector T cells that mediate type-1 inflammation [70]. The vitamin D analog TX527 promotes the surface expression of *CXCR6* on T-cells and inhibiting effector T cell reactivity while inducing regulatory T cell characteristics, promoting migration to sites of inflammation [71]. |
| HGFAC | calcium | | $4.0 \times 10^{-7}$ | Plays a role in converting hepatocyte growth factor (HGF) to its active form. Binding of HGF causes the up-regulation of *CXCR3*. *CXCR3* is preferentially expressed on $T_H1$ cells, while *CCR3* is expressed by $T_H2$ cells [72]. *CXCR3* binds the chemokine receptor *CCR3* and prevents an activation of $T_H2$-lymphocytes, biasing an immune response towards $T_H1$ inflammation [73]. *CXCR3* is able to increase intracellular $Ca^{2+}$ levels [74]. The D-test for calcium shows that the gene is located in the right tail ($p_D \simeq 5.9 \times 10^{-4}$). |
| DOK7 | | | $1.5 \times 10^{-6}$ | Activates *MuSK*, which is involved in concentrating *AChR* in the muscle membrane at the neuromuscular junction. The latter protein is critical for signaling between nerve and muscle cells, a necessity for movement, and is influenced by intracellular calcium [75]. Muscle weakness is a symptom of some severe COVID-19 patients [76]. The D-test for calcium shows that the gene is located in the right tail ($p_D \simeq 3.5 \times 10^{-5}$). |
| AGAP2 | vitamin D | + | $2.6 \times 10^{-6}$ | |
| OS9 | | | $3.8 \times 10^{-6}$ | |
| TSPAN31 | | | $4.4 \times 10^{-6}$ | |

weak signals may combine into a stronger signal if the directions of associations are consistent over the gene window. Such signals are invisible in the simple approach.

Fig 10 shows scatter plots for gene-wise log-transformed *p*-values for some of the GWAS we discussed above, namely, the severe COVID-19 GWAS against the M05B medication class and rheumatoid arthritis GWAS. Significance thresholds for the (simple) product-normal are also indicated. We note that the boundaries of the scattered *p*-value distributions appear to follow product-normal significance curves, justifying our choice to take the single-element product-normal to define significance thresholds for the combined $\chi^2$ statistic.

Fig 10A shows the co-analysis of COVID-19 and M05B GWAS signals. We observe that *CCR1* and *CCR3* are also Bonferroni significant under the simple meta-analysis test, however

**Table 3. Table of genes of significance ($p < 10^{-6}$) detected via ratio tests against COVID-19.** The column **E** lists the exposure, **O** the outcome, and **D** the test direction, with + for coherent and − for anti-coherent. We confirmed that all listed genes are $p_V$ significant under the exposure, but not the outcome.

| gene | E | O | D | $p_R$-value | COVID-19 context description |
|---|---|---|---|---|---|
| *KLC1* | vitamin D | COVID | + | $7.1 \times 10^{-7}$ | Kinesin-1 uncoats viral DNA and features in the COVID-19 virus-host protein interactions, belonging to the functional group of viral trafficking [77]. |
| *XRCC3* | | | | $2.7 \times 10^{-6}$ | |
| *ZFYVE21* | | | | $3.2 \times 10^{-6}$ | Regulates microtubule-induced PTK2/FAK1 dephosphorylation, which is important for integrin beta-1/ITGB1 cell surface expression. Integrins in host cells may play the role of alternative receptors to ACE2 for SARS-CoV-2 [43, 44]. |
| *RNF217* | calcium | | | $1.9 \times 10^{-6}$ | Member of the E3 ubiquitin-protein ligase family. A potential COVID-19 therapeutic pathway based on E3 ubiquitin ligases has been recently proposed in [78]. |
| *DDP9* | COVID | calcium | − | $5.8 \times 10^{-7}$ | Has been implicated before to be involved in the genetic mechanisms for critical illness due to COVID-19 [62]. |
| *HOXC4* | M05B | COVID | | $2.7 \times 10^{-6}$ | Related to an enhanced antibody response under the regulation of estrogens. Has been discussed before in the COVID-19 context in [79]. Potential confounding role of vitamin D ($p_V \simeq 0.06$). |
| *PPP2R3A* | calcium | | | $9.6 \times 10^{-6}$ | Interacts with *CDC6* [80], which is up-regulated at the early stages of human coronavirus 229E infection [81]. Potential confounding role of vitamin D ($p_V \simeq 0.03$). |

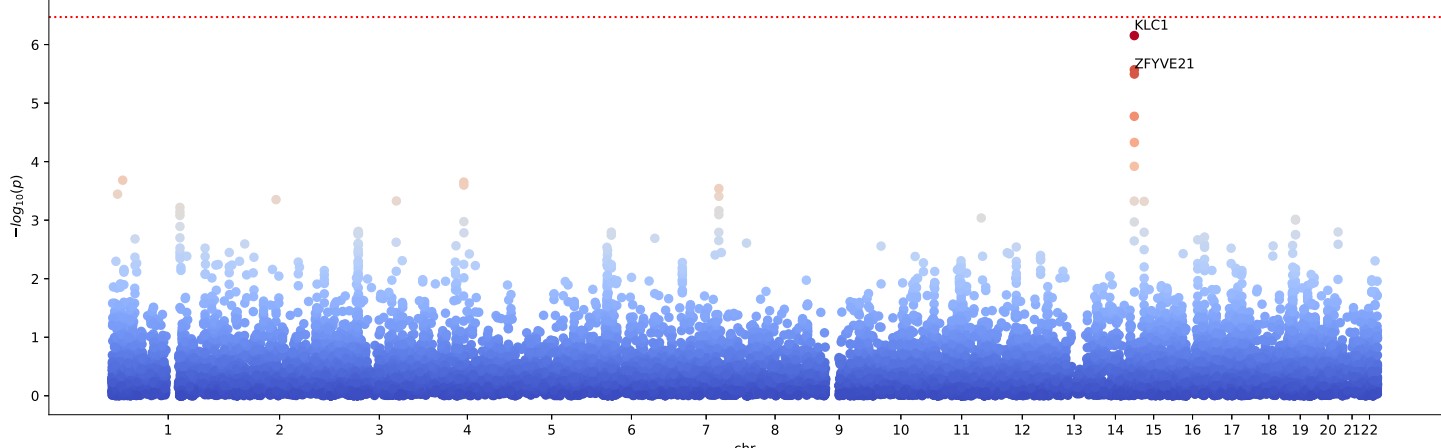

**Fig 9. Manhattan plot for ratio scoring Vitamin D concentration against severe COVID-19 in the coherent case, with ratio denominator given by COVID-19.** The Bonferroni significance threshold of $p = 3.4 \times 10^{-7}$ is indicated by the red dotted line (0.05 divided by the 18453 genes and 2 times 4 traits tested).

not *LZTFL1*. Also, while *CCR9* and *CXCR6* cross the $p < 10^{-6}$ threshold under the coherence test, not so under the simple single-element product-normal test. This confirms that the coherence test can improve the signal strength and may lead to fewer false negatives. Fig 10B shows severe COVID-19 against RA. While, again, all Bonferroni significant genes are also

**Table 4. Bonferroni significant pathways for the ratio test.** We tested against the 32284 gene sets of MSigDB 7.4. We list Bonferroni significant pathways ($p < 0.05/32284/7 \simeq 2.2 \times 10^{-7}$) and pathways close to significance.

| E | O | D | enriched pathway | # genes | *p*-value |
|---|---|---|---|---|---|
| RA | COVID | - | GOCC MHC protein complex | 9 | $3.6 \times 10^{-8}$ |
| | | | GOMF peptide antigen binding | 14 | $2.5 \times 10^{-8}$ |
| | | | KEGG allograft rejection | 22 | $8.1 \times 10^{-7}$ |
| | | | KEGG graft vs. host disease | 21 | $3.0 \times 10^{-7}$ |
| | | | Module 143 | 9 | $2.5 \times 10^{-7}$ |
| | | | Module 293 | 7 | $7.5 \times 10^{-7}$ |

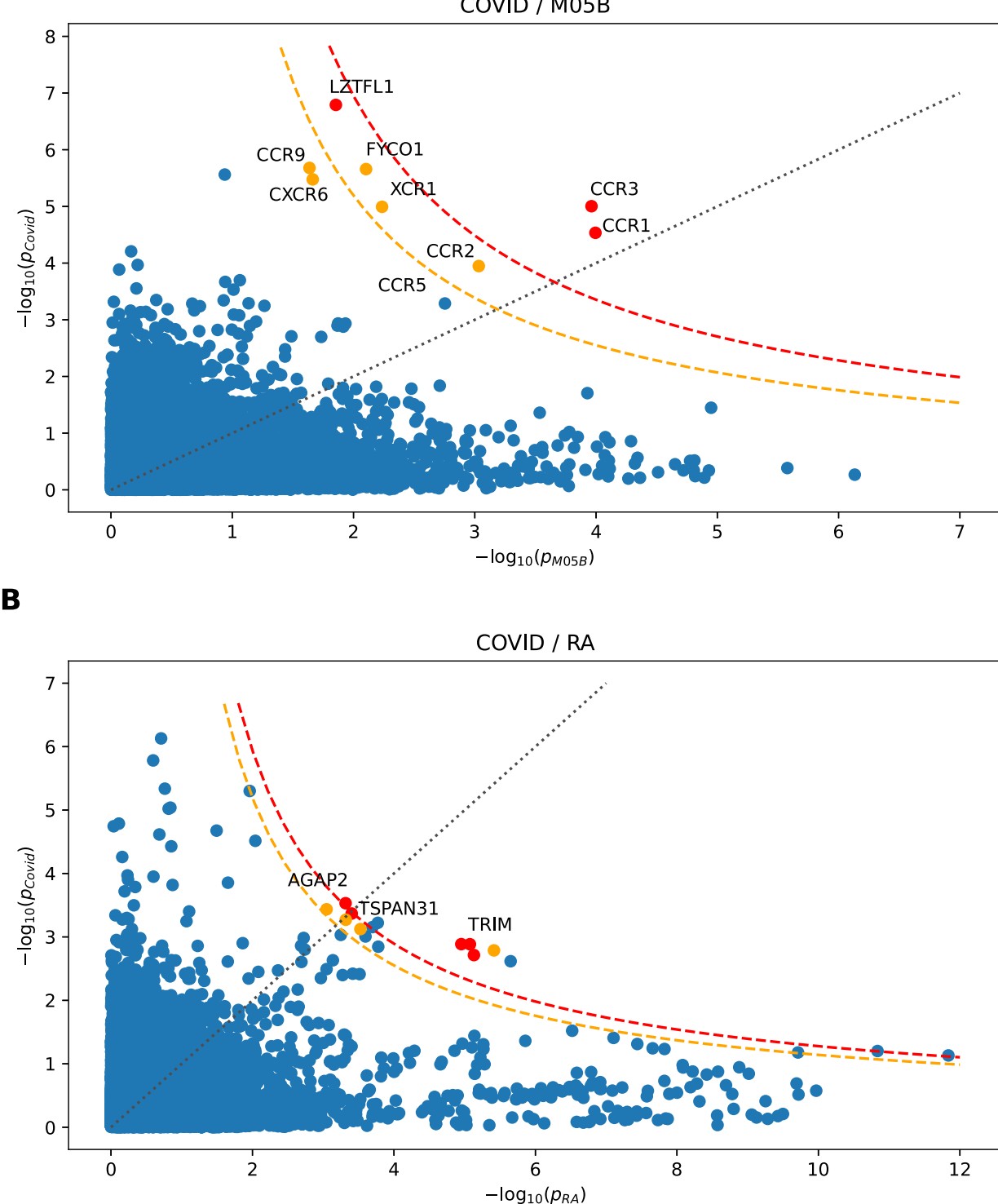

**Fig 10. GWAS gene scores, obtained separately for two GWAS using joint qq-normalization and standard $\chi^2$ based enrichment test, plotted against each other.** Each point corresponds to a gene. The gray dotted line marks the diagonal corresponding to equal $-\log_{10}$ $p$-values. The orange and red dashed curves mark the simple gene-wise product-normal threshold curves for significance of $p < 10^{-6}$ and Bonferroni significance, respectively. The color of a gene (orange and red) indicates the full SNP-wise cross scored $p$-value ($p < 10^{-6}$ and Bonferroni significant, respectively).

Bonferroni significant under the single-element test, we observe several additional genes which appear to be Bonferroni significant under the simple test. For instance, it appears that the single element test cannot completely resolve the TRIM gene cluster and hence may lead to false positives.

## 3 Discussion

The work presented here relies on the novel mathematical finding that the null distribution of the normal product of two effect size vectors with the same known covariance structure can be expressed as a weighted $\chi^2$ difference distribution (*cf.*, Eq (2)). Moreover, we showed that the ratio between this product and the square of one of the vectors—which allows for testing causal relationships—also relates to this distribution. In both cases the corresponding cdf and tail probabilities can be computed efficiently with Davies' algorithm.

This insight applies to GWAS because it enables testing for *coherent* SNP-wise effects within a gene window for two different traits. Importantly, our test is different from an additive test looking merely for co-significance. Indeed, testing whether the signs of the effects sizes tend to be identical (or opposite) within the window has potentially more power to identify genes that modulate both traits. Furthermore, our test is statistically rigorous, properly accounting for the SNPs' correlation structure (*i.e.*, LD), while still being able to be computed rapidly and accurately without any approximations. We have to assume the existence of an underlying joint normal distribution, and the possibility of jointly inferring the SNPs dependency pattern from a reference population.

We also introduced a simplified version of the coherence test, which essentially can be understood as a novel kind of product-normal based multiplicative meta-analysis of $\chi^2$ based gene scores. On a few specific examples, we observed that such a simpler test for co-significance may be sufficient for detecting highly significant genes. Nevertheless, we also observed that our gene-wise coherence statistic often provides added sensitivity and specificity, particularly in cases driven by several marginally significant genes.

We applied our method to identify genes with a role both in severe COVID-19 and medical conditions leading to the prescription of any of 23 medication classes. Our analysis revealed a strong signal of coherence between COVID-19 and M05B medications, with chemokine receptor genes *CCR1*, *CCR3* and *LZTFL1* as lead hits. The chemokine receptor of type 1 (*CCR1*) regulates bone mineralization and immune/inflammatory response. Mouse studies suggest that this gene plays a role in protection from inflammatory responses and host defense [27, 28]. The chemokine receptor *CCR3* is important for regulating eosinophils, leukocytes involved in many inflammatory pathologies [29]. In particular, mouse models suggest a complex role of *CCR3* in allergic diseases [30]. Due to their critical role in recruiting effector immune cells to the location of inflammation, chemokines are suspected to be a direct cause of acute respiratory disease syndrome, which is a major cause of death in severe COVID-19 (*cf.*, [31]). Hence, it appears plausible that the genes singled out by our new coherence test indeed play a role in both traits.

We also observed that genetic variants in genes related to both the adaptive (HLA) and innate immune system (TRIM genes) that are more frequent in subjects treated with medications indicated for specific autoimmune disorders tend to reduce the risk of severe COVID-19. Indeed, it is well known that autoimmune disorders are more common in females [32] who also have a smaller risk of severe COVID-19 compared to men [33]. While it is reasonable to expect that subjects with an increased risk for autoimmune disorders will tend to fight off infections more efficiently, the added value of our analysis is to pinpoint specific genes that are

potentially involved in mediating this effect and which may hint toward a protective pathway or therapeutic targets against severe COVID-19.

We then searched for coherence signals between COVID-19 and additional GWAS traits known to be related to diseases treated with M05B drugs. This analysis suggests that RA, vitamin D, and calcium concentration are traits of co-relevance for severe COVID-19. The RA trait provided further strong evidence for the relevance of *TRIM* genes of the innate immune system for a protective pathway, while vitamin D and calcium concentration traits implicated genes related to the differentiation between type-1 and type-2 immune responses. A possible explanation of the latter could be that many patients being prescribed class M05B medication suffer from osteoporosis. Vitamin D stimulates calcium absorption and is, therefore, often prescribed to these patients to increase their bone mineral density [34]. Data clearly support the function of vitamin D in bone growth and maintenance. However, evidence for a role of vitamin D in acute respiratory tract infections [35]—often observed for patients with severe COVID-19—is less clear cut: Vitamin D is thought to reduce the risk of infection, mainly due to factors involving physical barriers, natural cellular immunity and adaptive immunity [36]. Furthermore, low plasma vitamin D levels have been associated with the risk of infection [37]. However, direct conclusive evidence for its proposed protective function specific to severe COVID-19 is still lacking [38], even though possible links between the severity of COVID-19 and vitamin D are actively discussed in the current literature (see, for instance, [39–41] and references therein).

Interestingly, a similar mechanism of action via an influence on balancing between type-1 (inflammatory) or type-2 (anti-inflammatory) immune response, as for vitamin D concentration, can be associated with the top hit of our coherence analysis between COVID-19 and serum calcium levels, the Bonferroni significant gene *HGFAC*, *cf.*, Table 2. In general, we know that viruses also appropriate or interrupt $Ca^{2+}$ signaling pathways and dependent processes, *cf.*, [42]. In particular *HGFAC* up-regulates *CXCR3*, which binds the chemokine receptor *CCR3* and biases the immune response towards $T_H1$ inflammation. Note that vitamin D strongly increases the rate of calcium absorption, which may lead to suspect a confounding effect. However, we detect different coherent genes for the traits, suggesting that both calcium and vitamin D concentrations may independently influence the severity of COVID-19.

In addition, using the ratio test to perform a causal analysis, we detected a potential causal link from a genetic predisposition for vitamin D deficiency to the severity of COVID-19, mediated by the *ZFYVE21* gene. Note that *ZFYVE21* regulates microtubule-induced PTK2/FAK1 dephosphorylation, which is important for integrin beta-1/ITGB1 cell surface expression and thereby potentially impacts disease outcome by influencing alternative receptors to ACE2 for host cell entry, *cf.*, [43, 44].

A recent work using standard GWAS methods could not detect genetic evidence linking vitamin D to the severity of COVID-19 [45]. The discrepancy to our observed effects, which are only marginally below the Bonferroni threshold (*cf.*, Fig 9), is likely explained by the fact that our approach tested all genes individually, while in [45] the analysis was restricted to the set of SNPs proximal to known vitamin D pathway genes.

A novel aspect of our method is its capacity to identify candidate genes mediating the causal effects, but we cannot exclude potential confounders in this causal analysis. Our localized gene-centered approach was able to single out several plausible candidates for causal genes, which had already been discussed elsewhere in the COVID-19 context, and call for further investigations. This illustrates the power of the methods developed in this work as a discovery engine. Therefore, we believe that they will be useful for other studies trying to identify genetic players mediating pleiotropic effects. An expected increase in the power of severe COVID

GWAS and other relevant traits is likely to further refine the picture starting to emerge in our analysis.

In this work, we chose genes as a natural level of granularity to search for coherent effects, while our pathway analyses merely aggregate such gene-wise significance in coherence. Future work could extend the concept of coherence to entire pathways. Specifically, one can ask whether such groups of genes all tend to exhibit effects of the same sign. In fact, in our pathway analysis, we already implemented this for gene pairs whose SNPs are in strong LD and should therefore be analyzed jointly (so-called *meta-genes*, as introduced in the original Pascal approach [2]). It did not escape us that for multiple genes, one can compute the product normal of the signed aggregate gene effects introduced in (12). Future work may investigate whether such *pathway coherence* could provide an efficient means to study pleiotropy at this intermediate level and may have more power than gene-level and whole-genome coherence analysis. Similarly, our ratio-score in (3) could be extended to the pathway level, allowing Mendelian randomization at this level.

A limitation of our approach is that common genetic effects on different traits cannot be disentangled from potential other joint residual contributions. However, such contributions are likely to be negligible when using GWAS data from different populations. Nevertheless, even when co-analyzing data from GWAS with overlapping populations, it is possible to correct for the bias introduced by the phenotypic correlation (see section 4 in S1 Text for derivations and testing results).

Throughout this study, we assumed that the effects of sample size differences could be compensated by qq-normalization. Potentially, our method could profit from better ways to deal with GWAS pairs whose traits exhibit very different effect size distributions or sample sizes.

Finally, we can envisage other more general applications of the methods discussed in this paper. For instance, our approach could be used to correct for the auto-correlation structures in estimating the significance of correlations between time series. The technical results presented in this paper may therefore be of interest to other domains.

## 4 Methods

### 4.1 Linear combination of $\chi^2$ distributions

We denote the $\chi^2$-distribution with $n$ degrees of freedom as $[\chi_n^2]$. It is well known that the sum of $N$ independent $\chi_{n_i}^2$ distributed random variables $v_i$ is also $\chi^2$ distributed, *i.e.*,

$$\sum_i v_i \sim \left[\chi_{\sum_i n_i}^2\right].$$

However, no closed analytic expression is known for the distribution $\Xi$ of a general linear combination

$$\sum_i a_i v_i \sim \sum_i a_i [\chi_{n_i}^2] = [\Xi]\,, \tag{5}$$

where $a_i$ are real coefficients. Nevertheless, various numerical algorithms exist to compute the cdf of $\Xi$, denoted as $F_\Xi$, up to a desired precision. Perhaps most well known are Ruben's algorithm [46–48] and Davies' algorithm [11, 12]. The latter is the most relevant for this work, as it allows for negative $a_i$.

With $F_\Xi$ at hand, for a given real $x$ a right tail probability $p$ ($p$-value) can be calculated as

$$p = 1 - F_\Xi(x) \,. \tag{6}$$

## 4.2 Product-normal distribution

The product-normal distribution, the distribution of the product of two normal-distributed random variables $w$ and $z$, plays a central role in this work. The moment generating function for joint normal samples with correlation $\varrho$ reads [49]

$$M_{w,z}(v) = \frac{1}{\sqrt{(1 - (1 + \varrho)v)(1 + (1 - \varrho)v)}} \,.$$

Our key observation is that the above moment generating function factorizes into moment generating functions of the gamma distribution, $M_\Gamma(v|\alpha, \beta) = \frac{1}{(1 - \beta v)^\alpha}$, i.e.,

$$M_{w,z}(v) = M_\zeta(v|1/2, 1 + \varrho)\, M_{-\xi}(v|1/2, 1 - \varrho) \,.$$

Therefore,

$$zw \sim [\Gamma(1/2, 1 + \varrho)] - [\Gamma(1/2, 1 - \varrho)] \,. \tag{7}$$

For general parameters of the two gamma distributions, the corresponding difference distribution is known as the bilateral gamma distribution [50]. (Note that the subtraction in Eq (7) is in the distributional sense, so, even for $\varrho = 0$, the corresponding distribution does not vanish.)

Due to the well known relation between the gamma and $\chi^2$ distributions, one can also express the product-normal distribution in terms of the $\chi^2$ distribution introduced in the previous section. In detail,

$$zw \sim \frac{1 + \varrho}{2}[\chi_1^2] - \frac{1 - \varrho}{2}[\chi_1^2] \,. \tag{8}$$

The cdf of the product-normal can therefore be efficiently calculated using Davies' algorithm, as the distribution (8) is simply a linear combination of $\chi_1^2$ distributions. A similar relation can be derived for the product distribution of non-standardized Gaussian variables, albeit in terms of the non-central $\chi^2$ distribution, *cf.*, section 1 in S1 Text. Note that the relation (7) allows for a simple analytic derivation of a closed-form solution for the product-normal pdf, but not for the cdf. For completeness, details can be found in the section 2 in S1 Text.

## 4.3 Coherence test decorrelation

We make use of the eigenvalue decompositions $U_w \Sigma_w U_w^T = \Lambda_w$ and $U_z \Sigma_z U_z^T = \Lambda_z$, with $\Lambda_.$ the diagonal matrix of eigenvalues of $\Sigma_.$, to decorrelate the elements of each set. The index $I$ can then be written as

$$I = w^T z = w^T U_w^T U_w U_z^T U_z z = \hat{w}^T U_w U_z^T \hat{z} = \hat{w}^T K \hat{z} \,,$$

with $\hat{w} \sim \mathcal{N}(0, \Lambda_w), \hat{z} \sim \mathcal{N}(0, \Lambda_z)$ and $K := U_w U_z^T$. In components, $I$ reads

$$I = \sum_{i,j} K_{ij}\, \hat{w}_i \hat{z}_j \,.$$

For $\Sigma_w = \Sigma_z =: \Sigma$ the matrix $K$ is the identity matrix, and the result given in Eq (2) follows.

## 4.4 GWAS

As explained in the introduction, a prime example of very strong inter-element correlations are SNPs in LD. Recall that the univariate least squares estimates of the effect sizes $\beta$ in genome-wide association studies (GWAS) reads

$$\beta_i = \frac{1}{n} x_i^T y\,, \tag{9}$$

with $x_i$ the $i$th column of the genotype matrix $X$ of dimension $(n, p)$ and $y$ the phenotype vector of dimension $n$. Both $x$ and $y$ are mean-centered and standardized. $n$ is the number of samples and $p$ the number of SNPs. The central limit theorem and standardization ensures that for $n$ sufficiently large $z_i := \sqrt{n}\beta_i \sim \mathcal{N}(0, 1)$.

As a multi-variate model, we have

$$y = X\alpha + \epsilon\,,$$

with $\alpha$ the vector of $p$ true effect sizes and $\epsilon$ the $n$-dimensional vector of residuals with components assumed to be $\epsilon_i \sim \mathcal{N}(0, 1)$ and independent. Substituting the multi-variate model into (9), yields

$$\beta_i = \frac{1}{n} x_i^T (X\alpha + \epsilon)\,.$$

As a fixed effect size model, under the null assumption that $\alpha = 0$ (no effects), we infer that

$$z_i = \frac{1}{\sqrt{n}} x_i^T \epsilon\,.$$

We can stack an arbitrary collection of such $z_i$ to a vector $z$ via stacking the $x_i$ to a matrix $x$, such that

$$z = \frac{1}{\sqrt{n}} x^T \epsilon \sim \mathcal{N}(0, \Sigma)\,, \tag{10}$$

with $\Sigma := \frac{1}{n} x^T x$. Note that we made use of the affine transformation property of the multi-variate normal distribution.

It is important to be aware that $z$ is only a component-wise univariate estimation, and hence the null model (10) is for a collection of SNPs with effect sizes estimated via independent regressions.

We can also take the effects to be random variables themselves. Let us assume that independently $\alpha_i \sim \mathcal{N}(0, h^2/p)$, with $h$ referred to as *heritability*. We then have that

$$z_i = \frac{1}{\sqrt{n}} \left(x_i^T X\alpha + x_i^T \epsilon\right)\,,$$

such that

$$z \sim \mathcal{N}(0, h^2 L) + \mathcal{N}(0, \Sigma) = \mathcal{N}(0, h^2 L + \Sigma)\,, \tag{11}$$

with $L := \frac{1}{np} x^T X X^T x$, and where we assumed that $\alpha$ and $\epsilon$ are independent. Two remarks are in order. As $X$ runs over all SNPs, the calculation of $L$ usually requires an approximation, for instance, via a cutoff. Furthermore, the null model (11) requires an estimate of the heritability. Such an estimate can be obtained for via LD score regression [9].

### 4.5 Direction of association

The direction of effect of the aggregated gene SNPs can be estimated via the index

$$D := \sum_i z_i. \tag{12}$$

In detail, making use of the Cholesky decomposition $\Sigma = CC^T$ and the affine transform property of the multi-variate Gaussian, we have that as null

$$D \sim \mathcal{N}(0, |C|_F^2), \tag{13}$$

with $|.|_F$ the Frobenius norm. Testing for deviations from $D$ in the right or left tail indicates the direction of the aggregated effect size. Note that the (anti)-coherence test between pairs of GWAS introduced above is, alone, not sufficient to determine the direction for a GWAS pair but requires, in addition, testing at least one GWAS via (12) to determine the base direction of the aggregated gene effect. Furthermore, at least one GWAS needs to carry a sufficiently oriented signal in the gene such that (12) can succeed. We will also refer to testing for deviations from (13) as D-test. (Note that we regularize $\Sigma$ via thresholding eigenvalues smaller than $10^{-8}$ for the D-test.)

### 4.6 Ratio test derivation

Clearly, for $\Sigma_w = \Sigma_z = \Sigma$ we have that

$$\Pr(R \leq r) = \Pr(\hat{w}\hat{z} \leq r\hat{z}^2) = \Pr((\hat{w} - r\hat{z})\hat{z} \leq 0).$$

We define $\hat{v} = \hat{w} - r\hat{z}$ such that $\hat{v} \sim \mathcal{N}(0, (1 + r^2)\Lambda)$. Note that the component-wise correlation coefficient $\varrho$ between $\hat{v}$ and $\hat{z}$ reads

$$\varrho = -\frac{r}{\sqrt{1 + r^2}}.$$

Hence, from Eq (8) and section 1 in S1 Text, we deduce that

$$\hat{v}\hat{z} \sim \sum_i \frac{\lambda_i \sqrt{1 + r^2}(1 + \varrho)}{2}[\chi_1^2]$$

$$- \sum_i \frac{\lambda_i \sqrt{1 + r^2}(1 - \varrho)}{2}[\chi_1^2]. \tag{14}$$

We conclude that (4) holds.

### 4.7 Pathway enrichment

The gene scores resulting from the above coherence or ratio test can be utilized instead of the usual gene scores to perform a gene set (pathway) enrichment test. A pathway is thereby tested for enrichment in coherent or causal genes for a GWAS trait pair. We follow the pathway scoring methodology of [2]. Genes of a pathway close to each other are fused to so-called meta-genes, and coherence or ratio test-based gene scores are re-computed for the fused genes. The purpose of the fusion is to correct for dependencies between the gene scores due to LD. The resulting gene scores ($p$-values) are qq-normalized and inverse transformed to $\chi_1^2$ distributed random variables. This is followed by testing against the $\chi_n^2$ distribution, with $n$ the total number of (meta)-genes in the pathway. In this work, we tested for pathway enrichment against MSigDB 7.4 [51, 52].

### 4.8 SNP normalisation

As discussed in the Results section, the product normal combines evidence for coherent association in a multiplicative manner. A potential challenge to the proposed method arises when one of the two GWAS has associations with very low $p$-values. Such highly significant associations are common for GWAS with very large sample sizes. Without moderating these p-values, such associations may appear nominally co-significant as soon as the other GWAS provides a mild significance level. We propose two possible strategies to mitigate this.

One strategy is introducing a hard cutoff for very small SNP-wise $p$-values. The precise cutoff depends on the desired co-significance to achieve, and the amount of possible uplift of large $p$-values one finds acceptable. The dynamic is clear from Fig 1. If we target a co-significance of $p = 10^{-8}$ and accept to consider SNPs with a $p$-value of 0.05 or less in one GWAS to be sufficiently significant, we have to cut off $p$-values around $10^{-16}$. While such a cutoff ensures that no SNPs with $p$-values above 0.05 in one GWAS can become co-significant due to very high significance (i.e., $p$-value below $10^{-16}$) in the other GWAS, applying such a hard cutoff point hampers distinguishing differences in co-significance.

An alternative strategy is to transform all $p$-values, rather than only the most significant ones. For example, the so-called qq-normalisation, re-assigns uniformly distributed $p$-values according to the rank $r$, i.e. $p = (r + 1)/(N + 1)$, where $N$ is the number of $p$-values. (This approach implies the strongest $p$-value moderation and therefore is very conservative.) The product-normal statistic in (2) is then computed with $w$ and $z$ according to the inverse $\chi^2$ cdf of the respective $p$-values. Since for GWAS usually $N \simeq 10^6$, the most significant transformed $p$-value is $\sim 10^{-6}$. According to Fig 1 the other $p$-value has then to be smaller than $10^{-3}$–$10^{-4}$ to achieve (genome-wide) co-significance.

Since the qq-normalisation allows for combining two GWAS with significantly different signal strengths without the need to introduce an adhoc cutoff, it is our approach of choice and will be used in this work. We also prefer to apply this transformation for the ratio test in order to compensate for different signal strengths between the GWAS.

## Supporting information

**S1 Text. Cross-GWAS coherence test at the gene and pathway level: Supplementary Text with mathematical derivations.**
(PDF)

**S1 Table. Drug codes and corresponding classes for the drug GWASs of [16].**
(XLSX)

**S1 Fig. Manhattan plot showing the strong gene enrichment on chromosomes 3 and 12 for the severe COVID-19 GWAS.** The Bonferroni significance threshold (red dotted line) is taken to be $2.7 \times 10^{-6}$ (0.05 divided by 18453, the number of tested genes). Only a selection of significant genes is labeled.
(EPS)

**S2 Fig. QQ plot for coherent cross-scoring of COVID-19 against drug class M05B for different gene window sizes. Each data point corresponds to a gene. The European sub-population of the 1K Genome project has been used as a reference panel. The control factor $\lambda$ is taken to be the median of observed $-\log_{10}$ transformed $p$-values divided by $-\log_{10}(0.5)$.**
(EPS)

**S3 Fig. QQ plot for coherent cross-scoring of COVID-19 against drug class M05B.** Each data point corresponds to a gene. Reference panel and λ is defined as in S2 Fig.
(EPS)

**S4 Fig. Left: SNP *p*-values after rank transform for the *LZTFL1* gene.** The *x*-axis is numbered according to the *i*th SNP in the gene window ordered by increasing position. The dotted black lines indicate the transcription start and end positions (first and last SNP). Up bars correspond to M05B and down bars to severe COVID-19. Green indicates positive and violet negative association with the trait. Right: SNP-SNP correlation matrix inferred from the 1KG reference panel. Note that the gene contains at least three sizable LD blocks.
(EPS)

**S5 Fig. Left: SNP *p*-values after rank transform for the *TRIM26* gene under medication L04.** Annotation as in S4 Fig, but up bars corresponding to L04. Right: SNP-SNP correlation matrix inferred from the 1KG reference panel.
(EPS)

**S6 Fig. QQ plot for anti-coherent cross-scoring of COVID-19 against rheumatoid arthritis.** Caption otherwise as for S3 Fig.
(EPS)

**S7 Fig. QQ plot for ratio scoring Vitamin D concentration against severe COVID-19 in the coherent case, with ratio denominator given by COVID-19.** Caption otherwise as for S3 Fig.
(EPS)

## Acknowledgments

We like to thank A. L. Button, A. Brümmer, Z. Kutalik and S. O. Vela for valuable comments on an earlier draft of the manuscript.

## Author Contributions

**Conceptualization:** Daniel Krefl, Sven Bergmann.

**Data curation:** Daniel Krefl.

**Formal analysis:** Daniel Krefl.

**Funding acquisition:** Sven Bergmann.

**Investigation:** Daniel Krefl.

**Methodology:** Daniel Krefl, Sven Bergmann.

**Project administration:** Sven Bergmann.

**Resources:** Sven Bergmann.

**Software:** Daniel Krefl.

**Supervision:** Sven Bergmann.

**Validation:** Daniel Krefl.

**Visualization:** Daniel Krefl.

**Writing – original draft:** Daniel Krefl, Sven Bergmann.

**Writing – review & editing:** Daniel Krefl, Sven Bergmann.

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
