## [Decision Letter · Decision Letter 0]

8 Jun 2022

Dear Dr. Krefl,

Thank you very much for submitting your manuscript "Covariance of Interdependent Samples with Application to GWAS" for consideration at PLOS Computational Biology.

As with all papers reviewed by the journal, your manuscript was reviewed by members of the editorial board and by several independent reviewers. In light of the reviews (below this email), we would like to invite the resubmission of a significantly-revised version that takes into account the reviewers' comments.

Your manuscript has been evaluated, and the consensus is that it may merit recommendation for publication if issues raised in the reviewing phase by can be adequately addressed.

Please note that many such issues may take considerable effort, as they are mostly associated to ‘major revision’ request. If there remains any doubt that all the requests have been met, I reserve the right to have the revision re-reviewed.

We cannot make any decision about publication until we have seen the revised manuscript and your response to the reviewers' comments. Your revised manuscript is also likely to be sent to reviewers for further evaluation.

Sincerely,

Annalisa Barla

Guest Editor

PLOS Computational Biology

Ilya Ioshikhes

Deputy Editor

PLOS Computational Biology

Your manuscript has been evaluated, and the consensus is that it may merit recommendation for publication if issues raised in the reviewing phase by can be adequately addressed.

Please note that many such issues may take considerable effort, as they are mostly associated to ‘major revision’ request. If there remains any doubt that all the requests have been met, I reserve the right to have the revision re-reviewed.

Reviewer's Responses to Questions

**Comments to the Authors:**

Reviewer #1: This manuscript introduces a new method for estimating gene-level joint genetic effects simultaneously from two traits profiled through a pair of GWAS. This new method is based on testing against a null hypothesis defined by the product-normal distribution. The approach is very neat and the experimental results provided in the manuscript show that it works very well, and it is implemented as an open-source Python package freely available on GitHub. I do not see caveats or shortcomings in the presented work and therefore I can only congratulate the authors for their thorough work and give them the following very minor suggestions.

1. I do not think that the first paragraph of the introduction is the right motivating example, because when one thinks of a Pearson correlation calculated from independently drawn samples, we typically associate those samples with subject individuals and not with SNPs, which are the unit that are later defined not to be independent when in close proximity (pg. 3, second paragraph).

2. In page 4, when it says "Hence, even though the two traits share the same significant gene, they may not share the same functional mechanism", I think in the last bit you probably mean ".. they may not share the same genetic mechanism".

3. In page 5 you mention Davies' algorityhm for the first time, please include also there the citation to the paper.

4. In page 5, you define the distribution of the prodcut of two Gaussian random variables as "I ~ X(N, 1), with X being the variance-gamma distribution". However, X is a confusing choice of notation because first, it is more tipically used to denote a single numerical random variable and, second, you use it in page 26 to denote the genotype matrix. One straightforward solution could be to add VG as superscript to X.

5. Axis tickmarks and labels in figures 1, 2 and 4 to 10 are too small to be read comfortably.

6. In figure 2, please use panel letters instead of tagging "Top" and "Bottom".

Reviewer #2: The study by Krefl & Bergmann describes a new method for detecting genes with coherent association signals for two traits. They devised a new statistical test and applied it to extract the genetic overlap between COVID-19 severity and other traits. The authors propose that the method could be extended to assess causality through a MR-type of analysis. There are merits in the algorithm and the concept could be useful for the field.

Major comments:

1- Introduction: Overall, there are general problems in introducing concepts of the algorithms searching for genetic correlation between traits, also when introducing LD blocks, which have many issues as they typically contain variants that are not in strong LD with the reminder.

2-Lines 173-178: This property is very interesting. However, the manuscript does not clearly explore if the approach is overly conservative. The question of reducing false positives is briefly assessed and discussed in the context of a simulation study, but it would be interesting to show additional data about conservativeness and further discuss.

3-A comparison of results against existing methods of colocalization would be desirable, at least in the context of the main results with COVID-19.

4-In the real GWAS settings, how is significance threshold calculated when cross-scored against medications? In most situations, the authors only considered the number of genes for the penalty and not the number of different medications being tested. This should be indicated and be considered to draw conclusions.

5-What are the risks and potential biases of the results for cross-scoring across GWAS sharing individuals? This needs assessments or at least a discussion.

6-How would the test behave in situations where only one or a few SNPs from the gene are associated with strong p values? The example of COVID-19 for the two loci highlighted are not good examples of that because there are many variants associated in the two regions.

7-How is each gene defined to aggregate the pvalues for the test? This is not explained and is a central part of the methods for it to be adaptable and reproducible by others. Besides, this approach has obvious limitations in the context of complex traits as many of the GWAS hits lie outside genes and would entail a difficulty for adding or linking their effects to distant genes. Please, discuss.

8-Overall the causation analysis of severe COVID-19 linked to vitamin D deficiency should be downplayed unless compared with MR approaches as a sanity check.

9-What is the connection between higher likelihoods of M05B medication and severe COVID-19? If I interpreted this correctly (lines 267-268), the M05B medication likelihood should increase with age. Genetic risks for COVID-19 reduces with increasing age (as opposed to immune factors, comorbidities, etc). Thus, if both GWAS are referring to the same effect allele, we should be expecting an anti-coherence. Is that correct? Please, clarify this in the text as this is not obvious from the language used in the manuscript.

Minor comments:

English grammar needs to be improved.

Abstract: Th1 to Th2 immune reaction. What does it mean reaction in this context? Clarify or modify the sentence

Better define the severe COVID-19 phenotype in the text. There are many and different comparisons (A1, A2, C) in that consortium and should be make clearer which analysis is being used. It is not described in the main text nor in the SM.

Figure 5 and other Manhattan plots: please, add their corresponding QQ plots and lambdas as control. In the legend, it is not clear if correlation is referring to R2 and if the data for its calculation is for all the 1KG reference panel or for the particular population composition involved in the GWAS results. Clarify.

Figure 6 (also for figures 4 and 5 of the SM): Use of alternative colors for this figure of logP by position is desirable for better distinguishing positive and negative associations. Green and orange are not easy to distinguish.

Table 1: A clarification of the type of drugs the first column refers to will be helpful for the reader.

**Have the authors made all data and (if applicable) computational code underlying the findings in their manuscript fully available?**

Reviewer #1: Yes

Reviewer #2: Yes

PLOS authors have the option to publish the peer review history of their article (what does this mean?). If published, this will include your full peer review and any attached files.

Reviewer #1: No

Reviewer #2: **Yes: **Carlos Flores
---

## [Decision Letter · Decision Letter 1]

26 Aug 2022

Dear Dr. Krefl,

We are pleased to inform you that your manuscript 'Cross-GWAS coherence test at the gene and pathway level' has been provisionally accepted for publication in PLOS Computational Biology.

Best regards,

Annalisa Barla

Guest Editor

PLOS Computational Biology

Ilya Ioshikhes

Section Editor

PLOS Computational Biology

Dear Authors,

based on the comments I received through the second review process I recommend your paper for publication.

Reviewer's Responses to Questions

**Comments to the Authors:**

Reviewer #1: The authors have addressed all the suggestions I made and I'm satisfied with this revised version of the manuscript.

**Have the authors made all data and (if applicable) computational code underlying the findings in their manuscript fully available?**

Reviewer #1: Yes

PLOS authors have the option to publish the peer review history of their article (what does this mean?). If published, this will include your full peer review and any attached files.

Reviewer #1: No

---

## [Editor Report · Acceptance letter]

16 Sep 2022

PCOMPBIOL-D-21-02179R1 

Cross-GWAS coherence test at the gene and pathway level

Dear Dr Krefl,

I am pleased to inform you that your manuscript has been formally accepted for publication in PLOS Computational Biology. Your manuscript is now with our production department and you will be notified of the publication date in due course.

With kind regards,

Anita Estes
